# Detecting and Mitigating Indirect Stereotypes in Word Embeddings

## Abstract

Societal biases in the usage of words, including harmful stereotypes, are frequently learned by common word embedding methods. These biases manifest not only between a word and an explicit marker of its stereotype, but also between words that share related stereotypes. This latter phenomenon, sometimes called "indirect bias," has resisted prior attempts at debiasing. In this paper, we propose a novel method to mitigate indirect bias in distributional word embeddings by modifying biased relationships between words before embeddings are learned. This is done by considering how the co-occurrence probability of a given pair of words changes in the presence of words marking an attribute of bias, and using this to average out the effect of a bias attribute. To evaluate this method, we perform a series of common tests and demonstrate that measures of bias in the word embeddings are reduced in exchange for some reduction in the semantic quality of the embeddings. In addition, we conduct novel tests for measuring indirect stereotypes by extending the Word Embedding Association Test (WEAT) with new test sets for indirect binary gender stereotypes. With these tests, we demonstrate the presence of more subtle stereotypes not addressed by previous work. The proposed method is able to reduce the presence of some of these new stereotypes, serving as a crucial next step towards non-stereotyped word embeddings.

FIX

FIX

## 1 Introduction

Distributional word embeddings, such as Word2Vec (Mikolov et al., 2013a) and GloVe (Pennington et al., 2014), are computer representations of words as vectors in semantic space. These embeddings are popular because the geometry of the vectors corresponds to semantic and syntactic structure (Mikolov et al., 2013b). Unfortunately, societal stereotypes, such as those pertaining to race, gender, national origin, or sexuality, are typically reflected in word embeddings (Bolukbasi et al., 2016; Caliskan et al., 2017; Garg et al., 2018; Papakyriakopoulos et al., 2020). These stereotypes are so pervasive that they have proved resistant to many existing debiasing techniques (Gonen & Goldberg, 2019).

Techniques attempting to remove or mitigate bias in word vectors are common in the literature. The typical case study for bias mitigation methods in the literature is binary[1] gender. Subspace methods, such as hard debiasing from Bolukbasi et al. (2016) and GN-GloVe from Zhao et al. (2018b), attempt to identify or create a vector subspace of gender-related information (typically a "gender direction") and drop this subspace. Counterfactual Data Substitution from Maudslay et al. (2019), based on Counterfactual Data Augmentation from Lu et al. (2020), swaps explicitly gendered words to counter stereotyped associations. James & Alvarez-Melis (2019) and Qian et al. (2019) both propose methods to reduce bias towards binary gender by encouraging learned conditional probabilities of words appearing with "he" and with "she" to be equal.

Gonen & Goldberg (2019) showed that common "debiasing" methods failed to meaningfully reduce bias in word embeddings. They describe how bias can manifest not only as undesirable association

---

[1]We use the phrase "binary gender" to refer to the common yet unrealistic simplification of gender as just "male" or "female", which we take to be the main source of bias of study in this work. This is a limitation of this work.

between stereotyped words and words marking a bias attribute[2], but also between stereotyped words themselves. These manifestations are sometimes called *direct bias* and *indirect bias*[3], using the terminology introduced by Bolukbasi et al. (2016). An example of this second manifestation of bias is that the word "doctor" might be associated more strongly with stereotypically masculine[4] words than with stereotypically feminine words. At the time, bias mitigation algorithms commonly attempted to address direct bias while leaving indirect bias mostly present.

A common trend in the study of the indirect bias is the departure from *stereotypes* as the object of study in favor of *clustering*. While the measures introduced by Bolukbasi et al. (2016); Caliskan et al. (2017) attempt to quantify the existence of commonly understood stereotypes, work on indirect bias typically uses the measures introduced by Gonen & Goldberg (2019) which merely attempt to measure how well proposed bias mitigation methods disperse words with similar relationships to the bias attribute in the embedding space. These clustering measures, while useful at capturing some forms of indirect bias, are limited. In particular, it is unclear how dispersed stereotyped words *should* be in the embedding space, given that the stereotype of a word is not entirely arbitrary and can potentially be estimated based on its semantic, non-stereotypical, meaning.

These new bias measures have inspired countless new bias mitigation methods. Nearest neighbor bias mitigation from James & Alvarez-Melis (2019) attempts to equalize each word's association with its masculine (defined by the original undebiased embeddings) neighbors and its feminine neighbors. Double hard debias from Wang et al. (2020) projects off the direction defined by the "most gender-biased words" (again, based on alignment in the original embedding's "gender direction") in addition to the standard gender-related subspace. Bordia & Bowman (2019) modify the loss function when learning word embeddings to penalize neutral words having large components in the gender-related subspace which can then be dropped off. Kumar et al. (2020) propose RAN-Debias which attempts to disperse words in the embedding space that share similar binary gender biases (defined again by the original word embeddings) while preserving the original geometry as much as possible. Lauscher et al. (2020) describe multiple bias mitigation methods: the standard projection method, averaging original word vectors with an orthogonal transformation that attempts to swap the bias attribute, and a neural method that uses a loss function to group together words exhibiting a bias attribute away from neutral words. These methods, similarly to the bias measures of Gonen & Goldberg (2019), focus on the clustering and dispersion of words in relation to the bias attribute.

In current state-of-the-art models, word embeddings have largely been replaced by contextualized embeddings from transformer models such as BERT (Devlin et al., 2018) and GPT (Radford et al., 2018). However, word embeddings remain a popular object of study when quantifying bias in NLP, in part due to their simplicity and theoretical results that make them easier to reason about. As advances in the understanding of bias and stereotypes in word embeddings have been adapted for these newer models (Liang et al., 2020; May et al., 2019), novel techniques to measure and mitigate bias in word embeddings remain relevant.

## 2 BACKGROUND

### 2.1 WORD EMBEDDINGS

Many word embedding algorithms use the empirical probability that two given words appear near each other in the corpus (Levy & Goldberg, 2014; Pennington et al., 2014). This empirical probability is computed by counting how many times one word appears in the context of another as a *word–context* pair. A word–context pair is defined as a pair of words from the corpus that appear within a certain fixed distance from each other, the *window size*, and within the same sentence. A word–context pair designates one word as appearing in the context of another; in this paper, we will refer to a word–context pair as $(a, b)$ where $a$ is a the word appearing in the context of the

---

[2]We use the phrase "bias attribute" to refer to an attribute associated with stereotypes that are to be removed. The reader can assume "bias attribute" in this work will always refer to binary gender without loss of comprehension. We prefer "bias attribute" for generality.

[3]Or "explicit" and "implicit" bias. In this work we will refer to these as "direct bias" and "indirect bias" for clarity, even though they are actually just two different manifestations of the same phenomenon.

[4]Following the recommendations of Devinney et al. (2022), we prefer the terms "masculine" and "feminine" over "male" and "female" in this work.

word $b$. Contexts can be unidirectional or bidirectional. In the unidirectional case, in a word–context pair $(a, b)$, $a$ always occurs before $b$ in the corpus (or alternatively, always after). In the bidirectional case, for any pair of nearby words $a$ and $b$, there are two word–contexts pairs: $(a, b)$ and $(b, a)$.

From these corpus statistics, word embedding algorithms learn vectors for each word in a way that the word co-occurrence statistics can be derived from the geometry of the vectors. The exact details for how this is done is dependent on the exact word embedding algorithm used and is not important for this work.

## 2.2 Word Embedding Association Test

The Word Embedding Association Test (WEAT), introduced by Caliskan et al. (2017), is a common test used to quantify the presence of specific stereotypes in word embeddings. Given two sets of target words $\mathbb{X}$ and $\mathbb{Y}$ of equal size and two sets of attribute words $\mathbb{A}$ and $\mathbb{B}$, WEAT measures the association between the targets and the attributes. The sets are chosen so that $\mathbb{X}$ and $\mathbb{A}$ are stereotypically linked with each other, and similarly for $\mathbb{Y}$ and $\mathbb{B}$.

The association of a word $w \in \mathbb{X} \cup \mathbb{Y}$ with the attributes is

$$s(w, \mathbb{A}, \mathbb{B}) = \frac{1}{|\mathbb{A}|} \sum_{a \in \mathbb{A}} \cos(\vec{w}, \vec{a}) - \frac{1}{|\mathbb{B}|} \sum_{b \in \mathbb{B}} \cos(\vec{w}, \vec{b})$$

where $\vec{u}$ denotes the word vector corresponding to the word $u$ and

$$\cos(\vec{u}, \vec{v}) = \frac{\vec{u} \cdot \vec{v}}{\|\vec{u}\| \|\vec{v}\|}$$

is the cosine similarity between $\vec{u}$ and $\vec{v}$.

The outputs of WEAT are the test statistic, effect size, and $p$-value (of a permutation test), defined by the following equations respectively:

$$s(\mathbb{X}, \mathbb{Y}, \mathbb{A}, \mathbb{B}) = \sum_{x \in \mathbb{X}} s(x, \mathbb{A}, \mathbb{B}) - \sum_{y \in \mathbb{Y}} s(x, \mathbb{A}, \mathbb{B})$$

$$\text{EffectSize}(\mathbb{X}, \mathbb{Y}, \mathbb{A}, \mathbb{B}) = \frac{\frac{1}{|\mathbb{X}|} \sum_{x \in \mathbb{X}} s(x, \mathbb{A}, \mathbb{B}) - \frac{1}{|\mathbb{Y}|} \sum_{y \in \mathbb{Y}} s(x, \mathbb{A}, \mathbb{B})}{\text{stddev}_{w \in \mathbb{X} \cup \mathbb{Y}} s(w, \mathbb{A}, \mathbb{B})}$$

$$p = P(s(\mathbb{X}_i, \mathbb{Y}_i, \mathbb{A}, \mathbb{B}) > s(\mathbb{X}, \mathbb{Y}, \mathbb{A}, \mathbb{B}))$$

where $(\mathbb{X}_i, \mathbb{Y}_i)$ is a random partition of $\mathbb{X} \cup \mathbb{Y}$ into two sets of equal size.

Of these three measures, the effect size is most commonly used in the literature. In this work, we report the effect size along with the $p$-value. The effect size retains its original meaning from Caliskan et al. (2017) of measuring the strength and direction of the tested stereotype. We interpret the $p$-value not as a measure of statistical significance, as originally conceived. Instead, we interpret it as another measure of the tested stereotype, indicating how easily words in $\mathbb{X}$ can be separated from words in $\mathbb{Y}$ according to their association with $\mathbb{A}$ and $\mathbb{B}$: $p$-values of 0, 0.5, and 1 correspond to perfect separation according to the stereotype, no separation, and perfect separation according to the opposite stereotype, respectively. We will interpret these as relative measures: a WEAT effect size closer to zero and a WEAT $p$-value closer to 0.5 both suggest reduced presence of the stereotype in the embeddings.

## 3 Indirect Stereotypes

Previous work on measuring stereotypes has typically focused on measuring associations between stereotyped words and explicit markers of the stereotype, such as names[5] and semantically gendered words. In turn, previous work on mitigating bias have used these same markers as input for bias mitigation methods. By looking at the same sets of words for quantifying and mitigating bias, it

---

[5]While names do not inherently mark any bias attribute, aggregate groups of names typically isolate a bias attribute more directly than other groups of words can and, more importantly for this work, are commonly used as reference points in bias mitigation algorithms.

is easy to overestimate the effect of bias mitigation. This is what led Gonen & Goldberg (2019) to propose their new bias measures. These are based around measuring how clustered previously biased words are, but it is unclear what result should be desired from these measures. To address this issue, we demonstrate that we can capture some forms of the remaining bias as stereotypes, which we refer to as *indirect stereotypes*.

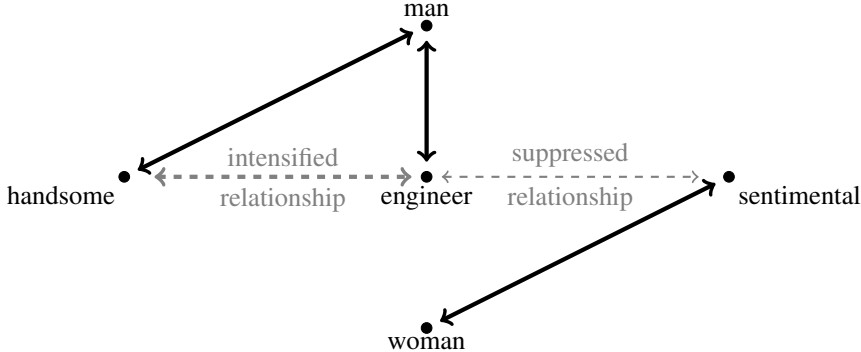

Figure 1: An example showing indirect stereotypes.

An example of an indirect stereotype is shown in Figure 1. There are three words here exhibiting a stereotype in regards to binary gender: "handsome" and "engineer," which are both masculine and not feminine, and "sentimental," which is feminine and not masculine. In a corpus with binary gender stereotypes, we would expect a quantifiably stronger association between "handsome" and "engineer" than between "sensitive" and "engineer" to exist purely because they are both stereotypically used to refer to men,[6] even if there is not a stereotype that engineers are more handsome than they are sensitive.

These stereotypes can come about as a result of sentences that exhibit multiple stereotypes at the same time. For example, the sentence "he is an a handsome engineer" exhibits the masculine stereotypes for "handsome" and "engineer". In a corpus that exhibits both of these stereotypes, this sentence would be more likely to occur than the following related sentences:



She is a handsome engineer.

He is a sensitive engineer.

She is a sensitive engineer.



This will result in "handsome" occurring with "engineer" more frequently than "sensitive" does. However, when word embeddings are "debiased" with respect to binary gender, the goal in previous work is typically to obtain word vectors that would predict each pair of the previous examples that differ only in choice of pronoun as equally likely to occur. This is not enough to correct the associations between "handsome," "engineer," and "sensitive." Furthermore, the association between "engineer" and "handsome" should exist even in sentences without explicit reference to binary gender, as in those cases "engineer" is more likely to refer to a man than a woman. More details on this analysis can be found in Appendix A.

To test indirect stereotypes, we use WEAT with the categories that are stereotyped according to binary gender (these are science/art, math/art, and career/family), as well a test set of professions and adjectives that are stereotypically masculine and feminine. These new test sets and the process used to generate them are detailed in Appendix B. While we will use WEAT as our primary test for stereotypes in word embeddings, subsequent work has called attention to shortcomings of the measure. We give a more thorough discussion with our analysis, at the end of Section 5.  NEW

## 4    BIAS MITIGATION METHOD

Instead of only looking at the co-occurrence probability of two words, we consider the probability a word co-occurs with another and one of the two also occurs near a word marking the bias attribute.

---

[6]Assuming there is no other relationship between "engineer" and "handsome" or "sensitive".

By using the word marking a bias attribute as a proxy for the bias attribute itself, we can determine how the relationship between the two words varies as the bias attribute varies. Then, we can approximate what the relationship between two words would be if one of them had no association with the bias attribute.

FIX

**Example 1** *Suppose we would like to mitigate one of the stereotypes in Figure 1, perhaps that "engineer" and "handsome" are more closely associated than they should be based on non-stereotypical semantics. In the corpus, it might be that an engineer is described as handsome three times as often when "engineer" refers to a man as opposed to when it refers to a woman. It might also be the case that "engineer" refers to a man twice as often as it refers to a woman. If "engineer" is adjusted so that it refers to men just as often as women, then the probability that an engineer is referred to as handsome should also be adjusted to be*

$$\frac{\frac{1}{2} \cdot P(\text{``handsome''}|man) + \frac{1}{2} \cdot P(\text{``handsome''}|woman)}{\frac{2}{3} \cdot P(\text{``handsome''}|man) + \frac{1}{3} \cdot P(\text{``handsome''}|woman)} = \frac{6}{7} \tag{1}$$

*of what it was before, at least when just considering occurrences where "engineer" refers to a man or woman. Whether "engineer" refers to a man or woman is not known to us a priori, but can be estimated based on whether "engineer" is near a word marking binary gender. Occurrences where "engineer" is not near such a word could be left as is or also be adjusted with this factor based on the reasoning in Section 3. In this work, we take the latter approach. We note that word embeddings do not attempt to quantify whether or not "handsome" describes an engineer, but rather if "handsome" occurs near "engineer". However, the same principle still applies.*

The general method is as follows. Suppose a word–context pair in the corpus is selected uniformly at random. Consider two words $a$ and $b$ where at least one of $a$ or $b$ is a word to be neutralized with respect to this bias. Additionally, let $\mathbb{X}$ and $\mathbb{Y}$ be two sets of words that are similar in usage but differ in terms of the bias attribute that is being mitigated. For example, in the case of binary gender, $\mathbb{X}$ could be the set $\{\text{``he''}, \text{``man''}, \dots\}$ and $\mathbb{Y}$ could be the set $\{\text{``she''}, \text{``woman''}, \dots\}$.

Define the following events for a random word–context pair $(c, d)$:

$$\text{A} : c = a, \qquad\qquad \text{X} : \text{the pair appears near a word in } \mathbb{X},$$
$$\text{B} : d = b, \qquad\qquad \text{Y} : \text{the pair appears near a word in } \mathbb{Y}.$$

A word–context pair is defined as being near a word in $\mathbb{X}$ or in $\mathbb{Y}$ if this word is within a certain distance from $c$ or from $d$. This distance can be taken to be the window size used for determining word–context pairs, for simplicity. In this work, we weight occurrences where a word in $\mathbb{X}$ or $\mathbb{Y}$ appears near just one of $c$ or $d$ half as much as if the word appeared near both of $c$ and $d$, but we could just as easily weight these occurrences the same.

The probability the word–context pair $(c, d)$ is $(a, b)$ can be decomposed as

$$P(\text{A} \cap \text{B}) = \frac{P(\text{B} \cap \text{X})P(\text{A}|\text{B} \cap \text{X}) + P(\text{B} \cap \text{Y})P(\text{A}|\text{B} \cap \text{Y}) - P(\text{A} \cap \text{B} \cap \text{X} \cap \text{Y})}{P(\text{X} \cup \text{Y}|\text{A} \cap \text{B})},$$

using the definition of conditional probability and inclusion-exclusion. This representation isolates the contribution of the bias attribute on $P(\text{A} \cap \text{B})$. The terms $P(\text{B} \cap \text{X})$ and $P(\text{B} \cap \text{Y})$ indicate the bias $b$ has, if any, while the terms $P(\text{A}|\text{B} \cap \text{X})$ and $P(\text{A}|\text{B} \cap \text{Y})$ describe how the relationship between $a$ and $b$ is influenced by the bias attribute.

Now if $b$ is replaced with a hypothetical neutral word $b'$ that has the same meaning except with no association to the bias attribute, the probability of $b'$ occurring with $a$ can be approximated. Denote by $\text{B}'$ the event that a randomly selected word–context pair $(c, d)$ satisfies $d = b'$.

The following exact equalities hold in this setting:

$$P((\text{X} \cup \text{Y}) \cap \text{B}) = P((\text{X} \cup \text{Y}) \cap \text{B}')$$
$$P(\text{X} \cup \text{Y}|\text{A} \cap \text{B}) = P(\text{X} \cup \text{Y}|\text{A} \cap \text{B}')$$
$$P(\text{X} \cap \text{B}') = P(\text{Y} \cap \text{B}').$$

The first two identities hold because $b$ and $b'$ have the same meaning except for the bias. The last identity holds because $b'$ has no bias.

In addition, the following relations are approximately true:

$$P(A|B \cap X) \approx P(A|B' \cap X)$$
$$P(A|B \cap Y) \approx P(A|B' \cap Y)$$
$$P(B \cap X \cap Y) \approx P(B' \cap X \cap Y)$$
$$P(A \cap B \cap X \cap Y) \approx P(A \cap B' \cap X \cap Y)$$

In the first two lines, the explicit presence of the bias attribute of interest is likely to overpower the bias $b$ has. In the last two lines, the approximate relationship is likely to hold as all the probabilities are small: the events are the intersection of three or more events and are subsets of $X \cap Y$.

These relations imply

$$P(X \cap B') = P(Y \cap B') \approx \frac{P(X \cap B) + P(Y \cap B)}{2}$$

and

$$P(A \cap B') \approx \frac{1}{P(X \cup Y|A \cap B)} \Big[ (P(A|B \cap X) + P(A|B \cap Y)) \cdot \tfrac{1}{2}(P(B \cap X) + P(B \cap Y)) \\ - P(A \cap B \cap X \cap Y) \Big]. \tag{2}$$

Reconsider Example 1. The factor in front is the original co-occurrence probability divided by the denominator of equation 1, with semantics replaced by word co-occurrence. In the example, $P(A \cap B \cap X \cap Y) = 0$ and $(P(B \cap X) + P(B \cap Y)) = 1$. With these substitutions, the remaining terms are the numerator of equation 1. With the switch from semantics to co-occurrence, this is now an approximation. **NEW**

A further approximation can be obtained by looking at $a$. Since $b'$ is neutral, if $a$ is replaced with a neutral word $a'$ and A replaced with a corresponding event $A'$, it holds that $P(A' \cap B') \approx P(A \cap B')$. A similar argument shows equation 2 holds when the role of $a$ and $b$ are switched. These two approximations can be averaged to yield a better approximation for $P(A' \cap B')$:

$$P(A' \cap B') \approx \frac{1}{P(X \cup Y|A \cap B)} \Big[ \tfrac{1}{4}(P(B \cap X) + P(B \cap Y))(P(A|B \cap X) + P(A|B \cap Y)) \\ + \tfrac{1}{4}(P(A \cap X) + P(A \cap Y))(P(B|A \cap X) + P(B|A \cap Y)) \\ - P(A \cap B \cap X \cap Y) \Big]. \tag{3}$$

This approximation is the basis of our proposed bias mitigation method. We propose the right hand side of equation 3 can replace the probability of a word–context pair being $(a, b)$ whenever this quantity would be used, where $a$ or $b$ is a word that is wanted to be neutralized. This can be done easily with word embedding methods that directly use the probability or counts of a specific word–context pair occurring, such as GloVe or PPMI factoring, but in principle could be adapted even to methods that only indirectly use these probabilities.

For all the terms in equation 3 to be defined, at least one occurrence each of $A \cap X$, $A \cap Y$, $B \cap X$, and $B \cap Y$ must be present in the corpus. Therefore, modified entries in the co-occurrence matrix must be limited to only those where both words appear in the corpus in a word–context pair satisfying X and in a pair satisfying Y. In practice this may not be a strong restriction. In a test case with binary gender, although only 34% of the words can have co-occurrences modified in this way, this includes most frequently occurring words. The vast majority of words that cannot be modified with this method have less than 100 occurrences in the corpus and would therefore have noisier embeddings. This is shown in Figure 2. Thus, words that cannot be applied equation 3 to can be dropped, yielding more pronounced bias mitigation. This is possible even with only eight words each in $\mathbb{X}$ and $\mathbb{Y}$. **NEW**

Of course, binary gender in English is particularly well-suited for this method, as words that mark binary gender are rather common (e.g., "he", "she", "man", "woman", etc.). For other types of biases, we would likely have to use less common words, such as names, resulting in fewer words we can apply equation 3 to. The entries of the original co-occurrence matrix with these words can be kept, although there will likely be less bias mitigation as a result.

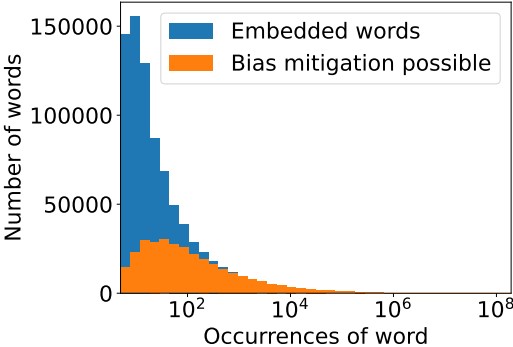

Figure 2: Number of embedded words versus number of occurrences of words in the corpus.

## 5 RESULTS

We conduct experiments by training GloVe embeddings on the UMBC webbase corpus (Han et al., 2013) and then attempting to mitigate the presence of bias from binary gender. We compare the proposed method from Section 4 with GloVe, with and without Counterfactual Data Subsitution (Maudslay et al., 2019). These are labeled as "Proposed", "Original", and "CDS" respectively in the following tables. Due to computational constraints, we only train CDS on 30 million sentences from the corpus. The results with training on the reduced corpus are shown on the left side of each table, while the results for the full corpus are shown on the right side of each table. In all tables, the best results for each half row are shown in bold.  NEW / FIX

For all methods, we train GloVe with an embedding dimension of 300. In determining the cooccurrence matrix, we use a window size of 15 and skip over words with less than 5 occurrences in the corpus. We also preprocess our corpus by removing capitalization, skipping sentences with less than five words, and using Stanford CoreNLP (Manning et al., 2014) to separate out certain lexemes from words (e.g., n't and 's).

For the proposed bias mitigation method, we use the following sets of words:

$$\mathbb{X} = \{\text{"he", "him", "his", "himself", "man", "men", "boy", "boys"}\}$$
$$\mathbb{Y} = \{\text{"she", "her", "hers", "herself", "woman", "women", "girl", "girls"}\}.$$

All results are obtained by training word vectors multiple times and reporting the median of the respective measures. Five sets each of word vectors are trained for the reduced corpus, while three sets are trained for the full corpus.

As a first test, we evaluate all three word embeddings on a series of common semantic tests (Jastrzebski et al., 2017). The relative change in the median results of these tests after the two tested bias mitigation methods are shown in Table 1. For all tests, a higher score indicates a better representation of the tested semantics in the word embeddings. The tests are grouped into three categories; from top to bottom, these are word similarity tasks, categorization tasks, and analogy tasks. Overall, our proposed method typically results in worse degradation of semantics than CDS does. This is most pronounced in the analogy tasks, where there is an approximately 15% decrease in accuracy in the smaller corpus. With the full corpus, there is less degradation in semantics. The analogy tasks still see an approximately 10% decrease in accuracy, but there is surprisingly a substantial *increase* in performance in the word similarity tasks and mixed results in classification.  NEW / FIX / NEW / NEW

The tasks that suffer with largest degradation in the full corpus (AP, BLESS, Battig, Google, MSR, and SemEval2012_2) are also the ones that see the largest improvement for standard GloVe when comparing the samller corpus to the larger. The degradation resulting from using GloVe trained on the smaller corpus for these tasks are $-6.48\%$, $-8.23\%$, $-10.63\%$, $-12.32\%$, $-9.67\%$, and $-6.99\%$. This performance is comparable to the performance of the bias mitigation method on the full corpus seen in Table 1. This suggests that this bias mitigation yield word vectors with semantic quality equivalent to training with a smaller corpus and therefore is most applicable when the training corpus is large enough that this is an acceptable trade-off.

Table 1: Semantic measures

| Test | Partial corpus | | Full |
| | Proposed | CDS | Proposed |
|---|---|---|---|
| MEN | $+\mathbf{0.48}\%$ | $-0.17\%$ | $+4.92\%$ |
| WS353 | $+\mathbf{0.82}\%$ | $-1.20\%$ | $+5.32\%$ |
| WS353R | $+\mathbf{0.45}\%$ | $-2.08\%$ | $+8.14\%$ |
| WS353S | $+\mathbf{0.90}\%$ | $+0.09\%$ | $+3.22\%$ |
| SimLex999 | $+\mathbf{0.23}\%$ | $+0.08\%$ | $+2.46\%$ |
| RW | $-10.45\%$ | $+\mathbf{0.71}\%$ | $+1.73\%$ |
| RG65 | $-10.34\%$ | $+\mathbf{1.14}\%$ | $-0.35\%$ |
| MTurk | $+\mathbf{1.07}\%$ | $-1.56\%$ | $+8.68\%$ |
| AP | $-7.76\%$ | $+\mathbf{1.22}\%$ | $-3.44\%$ |
| BLESS | $-8.97\%$ | $-\mathbf{0.64}\%$ | $-7.06\%$ |
| Battig | $-9.35\%$ | $-\mathbf{0.25}\%$ | $-7.22\%$ |
| ESSLI_2c | $+\mathbf{3.70}\%$ | $+\mathbf{3.70}\%$ | $+7.41\%$ |
| ESSLI_2b | $\pm\mathbf{0}\%$ | $\pm\mathbf{0}\%$ | $+6.90\%$ |
| ESSLI_1a | $\pm\mathbf{0}\%$ | $+\mathbf{2.86}\%$ | $\pm0\%$ |
| Google | $-23.41\%$ | $+\mathbf{0.15}\%$ | $-11.97\%$ |
| MSR | $-16.56\%$ | $-\mathbf{0.84}\%$ | $-10.44\%$ |
| SemEval2012_2 | $-8.93\%$ | $-\mathbf{1.58}\%$ | $-5.99\%$ |

Table 2: $p$-values (top) and effect sizes (bottom) for WEAT for unrelated biases

| Test | Partial corpus | | | Full corpus | |
| | Original | Proposed | CDS | Original | Proposed |
|---|---|---|---|---|---|
| Flower/Insect–Pleasantness | 2.9e−5 | 1.57e−4 | **1e−6** | 0 | 2.5e−5 |
| Music Instrument/Weapon–Pleasantness | **0** | **0** | **0** | **0** | **0** |
| Flower/Insect–Pleasantness | 1.067 | 1.065 | **1.261** | **1.339** | 1.187 |
| Music Instrument/Weapon–Pleasantness | 1.304 | **1.413** | 1.315 | 1.389 | **1.415** |

As another test for semantics, we compare all the word embeddings on two tests for WEAT that do not capture stereotypes related to binary gender. These tests exhibit the bias that music instruments are more pleasant than weapons, and flowers are more pleasant than insects. The results are shown in Table 2 for bot the $p$-values and effect sizes. Since these biases encode information that is not being attempted to remove, a small $p$-values and a large effect sizes are desirable. We see that all three word embeddings exhibit strong representations of these biases with only minor differences between them.

To test the standard direct stereotypes towards binary gender, we again use WEAT with three binary gender stereotypes used in the original paper. The $p$-values and effect sizes for these tests are in Table 3. All three of these stereotypes are present in the original GloVe embedding, with

Table 3: $p$-values (top) and effect sizes (bottom) for WEAT for direct stereotypes

| Test | Partial corpus | | | Full corpus | |
| | Original | Proposed | CDS | Original | Proposed |
|---|---|---|---|---|---|
| Math/Art–Masc./Fem. Words | 0.0388 | 0.202 | **0.638** | 0.0800 | **0.304** |
| Science/Art–Masc./Fem. Words | 0.0349 | 0.665 | **0.468** | 0.0723 | **0.586** |
| Masc./Fem. Names–Career/Home | 7.8e−5 | **1.55e−4** | 2.33e−4 | 7.8e−5 | **6.22e−4** |
| Math/Art–Masc./Fem. Words | 0.913 | 0.449 | **−0.203** | 0.734 | **0.286** |
| Science/Art–Masc./Fem. Words | 0.928 | −0.230 | **0.0436** | 0.761 | **−0.117** |
| Masc./Fem. Names–Career/Home | 1.761 | **1.548** | 1.592 | 1.774 | **1.428** |

Table 4: $p$-values (top) and effect sizes (bottom) for WEAT for indirect stereotypes

| Test | Partial corpus | | | Full corpus | |
|------|----------|----------|------|----------|----------|
|      | Original | Proposed | CDS  | Original | Proposed |
| Professions–Adjectives | 0.0289 | **0.0952** | 0.0408 | 0.00296 | **0.0532** |
| Math/Art–Adjectives | 0.00303 | **0.124** | 0.00249 | 6.99e−4 | **0.108** |
| Science/Art–Adjectives | 9.32e−4 | **0.00536** | 0.00155 | 0.00124 | **0.0236** |
| Career/Home–Adjectives | **7.8e−5** | **7.8e−5** | **7.8e−5** | **7.8e−5** | **7.8e−5** |
| Professions–Adjectives | 0.865 | **0.609** | 0.799 | 1.165 | **0.745** |
| Math/Art–Adjectives | 1.293 | **0.625** | 1.245 | 1.458 | **0.649** |
| Science/Art–Adjectives | 1.457 | **1.262** | 1.398 | 1.463 | **1.0133** |
| Career/Home–Adjectives | 1.842 | **1.773** | 1.838 | 1.825 | **1.774** |

the association between traditionally masculine and feminine names with career and home words being the strongest. The presence of all three of these stereotypes are reduced as a result of both bias mitigation methods, although the career/home stereotype is not substantially reduced by either method.

Lastly, we investigate the presence of indirect stereotypes as described in Section 3. We use a list of stereotypically masculine and feminine adjectives from Hosoda & Stone (2000) and look at its association with stereotypically masculine and feminine professions from Caliskan et al. (2017), as well as the same math/art, science/art, and career/home categories from the standard WEAT tests. The full word lists for our new tests are shown in Appendix B. The results of these experiments are in Table 4. From this, it can be seen that all these stereotypes do exist in the original word embeddings. Counterfactual Data Substitution does not succeed at significantly reducing any of these stereotypes. Our proposed method reduces the presence of all these stereotypes to a greater extent than CDS, although it is not uniformly successful. In particular, it has modest success in reducing the presence of the indirect stereotypes with professions and with science/art and is able to significantly reduce the presence of the math and art indirect stereotype.    **FIX**

These results are modest but suggestive. Even by our own measures, we are not able to fully mitigate all tested stereotypes present in the word embeddings. These measures themselves are not perfect. As shown by Ethayarajh et al. (2019); Schröder et al. (2021), the WEAT scores may incorrectly measure the presence of stereotypes. We prefer WEAT for our tests because it emphasizes aggregate stereotype associations over individual ones, but also include experiments with one alternative, Relational Inner Product Association (RIPA), in Appendix C. Another important consideration is that recent results have shown that intrinsic bias measures (including WEAT and RIPA) do not necessarily generalize to downstream tasks (Goldfarb-Tarrant et al., 2021; Orgad & Belinkov, 2022), so the performed tests alone cannot guarantee word embeddings lack bias. However, by considering stereotypes as a measure of indirect bias, these tests can be extended to downstream bias measures such as WinoBias (Zhao et al., 2018a) and WinoGender (Rudinger et al., 2018) more readily than the previously used dispersion measures.    **NEW**

## 6 DISCUSSION

In this paper, we discuss how indirect bias in word embeddings can manifest as stereotypes. Using the standard Word Embedding Association Test with additional test sets, we demonstrate that these "indirect stereotypes" have a substantial presence in word embeddings and are not removed by current debiasing methods. Furthermore, we propose a new method that attempts to directly mitigate these indirect stereotypes and demonstrate that this method can have some success in practice, albeit with trade-offs on the semantic quality of the resulting word vectors. This method is a first step towards a more thorough removal on indirect stereotypes. In particular, we demonstrate that indirect stereotypes can be mitigated even with only using direct markers of the bias they come from.    **NEW**

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

## A  ANALYSIS OF INDIRECT STEREOTYPE OCCURRENCE

Consider the following list of sentences:



He is a handsome engineer.

She is a handsome engineer.

He is a sensitive engineer.

She is a sensitive engineer.



The first sentence exhibits the indirect stereotype found in Figure 1 and is the most likely sentence to occur in a corpus that treats both "engineer" and "handsome" as masculine and non-feminine words and "sensitive" as a feminine and non-masculine word.

Let $P(A)$ for a list $A$ of words be the probability that a sentence in the corpus contains the words in $A$, given that it is one of the four example sentences above. As long as $P(\text{"he"}) > \frac{1}{2}$ and $P(\text{"handsome"}|\text{"he"}) > P(\text{"handsome"}|\text{"she"})$, then

$$P(\text{"handsome"}) = P(\text{"handsome"}|\text{"he"})P(\text{"he"}) + P(\text{"handsome"}|\text{"she"})P(\text{"she"})$$

$$> \frac{1}{2} \cdot P(\text{"handsome"}|\text{"he"}) + \frac{1}{2} \cdot P(\text{"handsome"}|\text{"she"}).$$

The expression on the last line is what we would expect the probability of a sentence with "handsome" appearing would be if "engineer" had no bias towards binary gender.

## B  INDIRECT STEREOTYPE WORD LISTS

For the list of adjectives, we use the "key" feminine and masculine traits from Hosoda & Stone (2000). We remove the words "feminine" and "masculine" from these lists as our goal is to quantify indirect stereotypes. We also remove "hard-headed" from the masculine list as it it outside of the model's vocabulary.

Feminine adjectives: affectionate, sensitive, appreciative, sentimental, sympathetic, nagging, fussy, emotional

Masculine adjectives: handsome, aggressive, tough, courageous, strong, forceful, arrogant, egotistical, boastful, dominant

For the list of professions, we use the list from Caliskan et al. (2017) and take the ten jobs with the highest proportion of women and of men according to the data that they derive from the Bureau of Labor Statistics.

Feminine professions: therapist, planner, librarian, paralegal, nurse, receptionist, hairdresser, nutritionist, hygienist, pathologist

Masculine professions: plumber, mechanic, carpenter, electrician, machinist, engineer, programmer, architect, officer, paramedic

## C  RIPA TESTS

NEW

To test the bias of individual words, we use Relational Inner Product Association (RIPA) from Ethayarajh et al. (2019). We test the same words use in the WEAT tests for direct biases from Section 5,

Table 5: RIPA scores

| Word | Partial corpus | | | Full corpus | |
|------|----------|----------|--------|----------|----------|
| | Original | Proposed | CDS | Original | Proposed |
| executive | $-5.63$ | $-3.27$ | $\mathbf{-0.70}$ | $-5.60$ | $\mathbf{-3.44}$ |
| management | $-5.56$ | $-4.20$ | $\mathbf{-1.93}$ | $\mathbf{-3.55}$ | $-3.91$ |
| professional | $\mathbf{-0.75}$ | $-2.91$ | $-1.00$ | $\mathbf{-0.90}$ | $-1.22$ |
| corporation | $-3.63$ | $\mathbf{-2.21}$ | $-2.84$ | $-4.11$ | $\mathbf{-1.84}$ |
| salary | $-2.36$ | $-3.28$ | $\mathbf{-1.62}$ | $\mathbf{-2.83}$ | $-3.85$ |
| office | $-4.01$ | $-1.40$ | $\mathbf{-1.02}$ | $-3.04$ | $\mathbf{-2.04}$ |
| business | $-5.31$ | $-3.20$ | $\mathbf{-1.01}$ | $-4.48$ | $\mathbf{-2.74}$ |
| career | $-1.92$ | $\mathbf{-1.16}$ | $-2.59$ | $-4.13$ | $\mathbf{-2.45}$ |
| home | $\mathbf{1.16}$ | $-1.94$ | $-2.66$ | $\mathbf{-0.15}$ | $-1.31$ |
| parents | $3.13$ | $\mathbf{-0.59}$ | $-0.82$ | $4.45$ | $-0.88$ |
| children | $1.97$ | $-3.95$ | $\mathbf{-0.88}$ | $3.82$ | $\mathbf{-2.87}$ |
| family | $\mathbf{-0.06}$ | $-2.11$ | $-1.97$ | $\mathbf{-1.47}$ | $-2.53$ |
| cousins | $1.01$ | $0.67$ | $\mathbf{0.21}$ | $0.27$ | $0.28$ |
| marriage | $3.40$ | $-1.81$ | $\mathbf{1.60}$ | $4.13$ | $-1.89$ |
| wedding | $5.66$ | $1.81$ | $\mathbf{1.34}$ | $6.89$ | $\mathbf{3.07}$ |
| relatives | $\mathbf{-0.50}$ | $-1.00$ | $-1.54$ | $\mathbf{-0.77}$ | $-0.82$ |
| science | $-3.22$ | $-3.50$ | $\mathbf{-1.81}$ | $-2.67$ | $\mathbf{-2.00}$ |
| technology | $-3.51$ | $-2.82$ | $\mathbf{-0.75}$ | $-3.13$ | $\mathbf{-0.74}$ |
| physics | $-5.23$ | $-1.79$ | $\mathbf{-1.25}$ | $-2.93$ | $\mathbf{-0.59}$ |
| chemistry | $\mathbf{-0.56}$ | $0.57$ | $-1.39$ | $-2.99$ | $\mathbf{-0.89}$ |
| nasa | $-3.05$ | $\mathbf{0.25}$ | $-0.53$ | $-3.40$ | $\mathbf{0.90}$ |
| experiment | $-1.66$ | $\mathbf{0.91}$ | $-1.16$ | $-2.04$ | $\mathbf{0.49}$ |
| astronomy | $-2.20$ | $-1.53$ | $\mathbf{0.02}$ | $-3.46$ | $-1.55$ |
| math | $1.96$ | $-1.01$ | $\mathbf{-0.40}$ | $2.13$ | $-0.54$ |
| algebra | $-1.43$ | $\mathbf{-0.26}$ | $1.83$ | $\mathbf{-0.57}$ | $-1.54$ |
| geometry | $-2.65$ | $\mathbf{1.10}$ | $1.91$ | $\mathbf{-2.05}$ | $3.01$ |
| calculus | $-4.25$ | $-2.19$ | $\mathbf{0.09}$ | $-5.73$ | $-1.74$ |
| equations | $-2.83$ | $-0.87$ | $\mathbf{0.59}$ | $-2.57$ | $\mathbf{-0.27}$ |
| computation | $-2.10$ | $-0.30$ | $\mathbf{-0.20}$ | $-2.02$ | $\mathbf{-0.23}$ |
| numbers | $-2.88$ | $\mathbf{-1.70}$ | $-1.76$ | $-3.69$ | $\mathbf{-2.27}$ |
| addition | $-1.42$ | $\mathbf{-0.96}$ | $-1.73$ | $-1.98$ | $\mathbf{-1.59}$ |
| dance | $4.10$ | $\mathbf{-0.00}$ | $-2.51$ | $4.84$ | $\mathbf{1.34}$ |
| literature | $-0.73$ | $-2.30$ | $\mathbf{-0.23}$ | $0.54$ | $-1.60$ |
| poetry | $1.21$ | $1.01$ | $\mathbf{-0.89}$ | $1.83$ | $2.36$ |
| novel | $-0.91$ | $\mathbf{-0.37}$ | $0.74$ | $-0.37$ | $\mathbf{-0.27}$ |
| symphony | $\mathbf{0.01}$ | $2.36$ | $0.60$ | $-3.52$ | $\mathbf{1.31}$ |
| art | $-3.54$ | $\mathbf{-1.74}$ | $-2.06$ | $-1.46$ | $\mathbf{-0.62}$ |
| sculpture | $1.53$ | $0.87$ | $\mathbf{0.67}$ | $0.35$ | $0.79$ |
| drama | $\mathbf{0.22}$ | $0.99$ | $-0.62$ | $0.55$ | $\mathbf{-0.30}$ |

except the words "Einstein" and "Shakespeare" as a gender association for these words would not be stereotypical. For each word, we average the RIPA score for all pairs of feminine and masculine words tested by WEAT to report a single score. A positive score means a feminine association while a negative score means a masculine association. The results are shown in Table 5. Neither our proposed method nor CDS consistently outperform the other at producing words with no binary gender bias, although it appears that CDS produces the least gendered word vectors slightly more often. The new bias mitigation method is able to consistently reduce the gender association of the tested words and rarely produces word with a stronger association with binary gender than the original vectors did.

We do not report RIPA scores for indirect bias. This is for two reasons. First, RIPA requires words marking the bias attribute to be paired. For direct binary gender stereotypes, there is usually a natural choice for this pairing. For the indirect stereotypes we test, there is not. Since the choice of pairing can have significant effects on the scores, this limits the ability of RIPA to measure stereotypes in this situation. Furthermore, RIPA emphasizes the individual stereotype of words. While this is important in the case of direct stereotypes, for indirect stereotypes the relationship is only undesirable in aggregate. As an example, if the word vector for "carpenter" is more closely related with the vector for "strong" than "emotional", this isn't necessarily because of binary gender stereotypes.

