# OpenReview forum: "Detecting and Mitigating Indirect Stereotypes in Word Embeddings"
_ICLR.cc/2023/Conference — Submitted to ICLR 2023_

### Official Review · Reviewer_T3X4 · 2022-10-24

**Confidence:** 4
**Correctness:** 3
**Technical Novelty And Significance:** 3
**Empirical Novelty And Significance:** 2
**Recommendation:** 5

**Clarity, Quality, Novelty And Reproducibility:**

- The idea is novel - the authors have identified a valid source of bias and provided a technique to mitigate it.
- The new co-occurrence probability value is straightforward to compute and implement so reproducibility is not an issue.
- No additional resources needed beyond existing word-lists (for at least the types of bias addressed in the work).

**Strength And Weaknesses:**

Strengths: The paper does introduce a good line of thought about indirect biases. Recent work analyzing WEAT has highlighted similar shortcomings and so this is a welcome idea. The substitution suggested is straightforward to implement.

These are the weaknesses I have identified:
- The empirical justification for this task is weak. For instance, WEAT is treated as a gold-standard and the effect score is treated as evidence. WEAT scores themselves are highly sensitive to the word-lists being used and recent results call into question using WEAT directly [1]. It could be that we only get semantic degradation but no additional improvements over CDS with this method. Ideally we like to see raw co-occurrence statistics or something decoupled from the metric itself.
- There seems to be a non-trivial semantic degradation compared to CDS (unless I am misunderstanding the results section).
- From the paper: "Therefore, the vocabulary must be limited so that all words in the corpus occur in a word–context pair satisfying X and in a pair satisfying Y." - This in particular seems like a strong imposition (I know the paper claims otherwise). The benefit of methods that try to identify and drop subspaces is that we can work with non-comprehensive word-lists. I am curious if this doesn't work until a fully fleshed out word-list for the embeddings is developed. I am happy to be corrected on this since it feels to me like a huge chunk of the vocabulary would need to be dropped if we are unable to acquire a large word-list (say for addressing a second form of bias beyond gender like caste or race).

[1]: https://aclanthology.org/P19-1166/


**Summary Of The Paper:**

The paper introduces introduces a technique to mitigate indirect bias. To use an example from the paper, after debiasing sentences [ "he is a handsome engineer", "she is a sensitive engineer" ] and removing the gendered pronoun, the indirect biases ("handsome engineer" and "sensitive engineer") still persist. Empirical motivation for addressing these indirect biases is provided by showing that implementing this co-occurrence probability value substitution produces a more pronounced debiasing effect.

The solution recommended is substituting the entry for a word-pair (a,b) in the word co-occurrence matrix with the formula from equation 2.

Overall we get mixed results at de-biasing with degradation in semantics - the abstract claims no or minimal degradation but the results section seems to show a significant (at least from what I can see) hit compared to using CDS.

**Summary Of The Review:**

- Overall, a good idea and a well-identified problem and a reasonable attempt at addressing it.
- The focus of the paper is an area of significant social import.
- There are some mild empirical concerns - mixed results, semantic degradation and a few issues with how the problem is motivated.

---

> ### Author Response · Authors · 2022-11-17
> **Response**
>
> 1. We thank the reviewer for calling attention to the important concerns and related work around debiasing measures. We now report the RIPA scores for the test words we use for WEAT direct bias tests.  We recognize the shortcomings of WEAT but also contend it's the most appropriate measure for the indirect bias tests.  An immediate issue with attempting to use RIPA on the indirect bias tests is that RIPA assumes the words marking bias attributes are paired.  In our experiments, the RIPA scores can be quite sensitive to the choice of pairing.  As there's not a natural choice of pairing for the indirect bias test cases, this severely limits our ability to interpret the RIPA scores for these cases.
>
>     A more significant consideration is that with indirect stereotypes the main concern is the aggregate association and not the individual ones. For example, it's not necessarily problematic that the word "carpenter'' has an association with the word "strong''.  This association alone could result in ``carpenter'' having a larger association with masculine adjectives than feminine ones.  It's only problematic if this trend holds so consistently that the stereotype of a word is recoverable from its association with other stereotyped words.  This is what the WEAT scores attempt to measure, especially the $p$-value.  We've added a brief discussion commenting on these drawbacks and introducing RIPA in Section 5, with the RIPA results in Appendix C.
> 2. We elaborate more on the semantic degradation in Section 5 with further experiments.  In short, this degradation appears to be less significant with more training data and seems to be equivalent to training word vectors on a smaller corpus. In particular, this is now in Table 1 and the surrounding discussion.  We have also revised the abstract in light of this semantic degradation.
> 3. We elaborate more on the vocabulary dropping issue.  For the small test set of eight masculine and eight feminine words, we were able to apply equation (2) to 34\% of the vocabulary.  This is still a large vocabulary list, so it is reasonable to simply drop the rest.  We agree that it would be difficult to find a similarly small set of words for other types of biases.  However, it's possible to just use equation (2) whenever it is applicable and leave the remaining co-occurrence probabilities the same. In our own experiments we find that the semantic performance is very similar to original GloVe but with less mitigation of stereotypes.  We've added some comments about this at the end of Section 4.

---

### Official Review · Reviewer_u8Li · 2022-10-26

**Confidence:** 4
**Correctness:** 4
**Technical Novelty And Significance:** 2
**Empirical Novelty And Significance:** 2
**Recommendation:** 5

**Clarity, Quality, Novelty And Reproducibility:**

The approach for mitigating bias is novel but I am skeptical about its impact. The problem also focuses only on gender-occupation bias which is justified since they wanted to evaluate the impact of the approach but notes on whether or not it can extended for biases other than gender would be a good contribution. The method seems generic but more details on the extension will be helpful. I also felt that the authors can be a bit more descriptive while explaining the methodology through examples.

**Strength And Weaknesses:**

Strength:
The paper targets an important problem of mitigating indirect bias in word embeddings. They propose a novel approach to modify relationships between co-occurring words before the embeddings are learned.

Weakness:

I found the details of the methodology slightly lacking. The experiment details have been briefly explained in the appendix which can be included in the main paper. Additionally, the math has been explained but the visualization of the approach through examples would have benefited the reader. The improvements are not uniform so details on which cases are positively/negatively impacted by the approach would have helped understand the impact of the approach.

**Summary Of The Paper:**

Authors discuss the evaluation of indirect bias/stereotypes in word embeddings and propose a methodology to mitigate them. They do this by modifying the relationships between words before the embeddings are learned. They compare their results with CDS method (Counterfactual Data Substitution) and find that their proposed method is better than CDS in some cases. The improvement, however, is not uniform and there are cases (like the analogy task) where the accuracy is reduced by around 20%. They also calculate p values and effect sizes for undirect bias to evaluate if this approach also removes biases that are not being attempted to remove (e.g Flower/Insect–Pleasant/Unpleasant)

**Summary Of The Review:**

The paper is well motivated but lacks in-depth analysis to actually evaluate the impact of the approach. It will benefit from a more detailed explaination of the approach and additional analysis/experiments to understand which cases are positively impacted and why.

---

> ### Author Response · Authors · 2022-11-17
> **Response**
>
> 1. We thank the reviewer for pointing out areas for clarification. The experiment details have been added to the beginning of Section 5.
> 2. To help illustrate the method, we have now emphasized the example at the beginning of Section 4 and refer back to it just before the end of the derivation of formula (2), carrying the example all the way through the methodology for visualization.
> 3.  We have conducted more thorough experiments to demonstrate the method's results are more consistent than originally described.  In addition, we offer an explanation for why certain semantic tests have worse performance. These results and discussions are now in Section 5, and in particular Table 1.
> 4. We further discuss the challenges in extending this method beyond binary gender at the end of Section 4.

---

### Official Review · Reviewer_GLiH · 2022-10-31

**Confidence:** 3
**Correctness:** 3
**Technical Novelty And Significance:** 3
**Empirical Novelty And Significance:** 3
**Recommendation:** 6

**Clarity, Quality, Novelty And Reproducibility:**

* The paper is clear and well written, and introduces a simple but novel method for mitigating indirect biases through a replacement of sensitive pairs.
* There is no public code and few examples of replacements are given, but for the most part the paper is reproducible.

**Strength And Weaknesses:**

* This paper tackles a relatively novel area, the systematic study and mitigation of implicit bias in word embeddings, which is an important area due to the wide use of such embeddings in a variety of applications, and the potential harm associated with the presence of implicit biases.
* The method introduced is novel and a useful step towards more robust mitigation of indirect biases in word embeddings and language models.
* The mitigation method introduced in this paper is quite basic and it is unclear if it would be possible to extend it to other domains with less well defined characterizing words (in this study the authors use "he","him","his"..../"she","her","hers",..., but I don't think such simple definition would be possible for most marginalized demographics.
* From the experiments section, it is clear that further study is needed to successfully mitigate indirect biases across models.

**Summary Of The Paper:**

This paper tackles the problem of indirect bias. The authors define how indirect bias in a corpus can be quantified, then introduce a method to reweight adjectives to correct their frequency in association with gendered nouns. These methods are then tested on a variety of embeddings, showing marginal improvements in some specific cases.

**Summary Of The Review:**

This is an interesting study of indirect stereotypes in word embeddings, an important issue for language models. This paper provides an interesting first step towards gaining an understanding on building fairer and less biases models. While the mitigation procedure is relatively constrained in its application and provides only modest improvements, it is a useful step in the right direction.

---

> ### Author Response · Authors · 2022-11-17
> **Response**
>
> 1. We thank the reviewer for their positive feedback and for raising questions about generalizing the method. We agree that implementing this method for marginalized demographics other than binary gender is tricky, as is often the case in other bias mitigation methods.  As discussed in the paper, names are often used in the literature to test for biases such as race and ethnicity.  In principle, the reference sets $X$ and $Y$ could both be sets of names.  These will likely be less common than the words we use for binary gender.  The straightforward solution is to work with these words and use equation (2) whenever it is applicable and use the original co-occurrence probabilities when it is not, although of course this solution is less than ideal.  We added a discussion at the end of Section 4 to this effect, and agree it is an important direction.
> 2. Code was submitted with the supplementary materials for this submission. If accepted, we intend to also upload this code to GitHub and will provide a link in the paper.

---

### Author Response · Authors · 2022-11-17
**Revision**

We thank the reviewers for their valuable and thoughtful feedback. We have submitted a revised version of the manuscript with changes tagged and highlighted. We use the tag "FIX" for changes and "NEW" for additions.

---

### Decision · Program_Chairs · 2023-01-20

**Decision:**

Reject

**Justification For Why Not Higher Score:**

As pointed out in the meta-review, we have considered this paper from very different aspects, by taking into account the paper content, the authors' responses, as well as reviewers' questions. By weighing the strengths of this work and its weaknesses (mainly around the constrained technical mitigation method, mixed results, and its extension for biases other than gender), we do not recommend a higher score at this moment and would like to encourage the authors to incorporate reviewers' suggestions to further improve this work.

**Justification For Why Not Lower Score:**

N/A

**Metareview: Summary, Strengths And Weaknesses:**

This paper looks at a novel aspect of biases (indirect stereotypes) in word embeddings and introduces a new approach to modify the biased relationships between words before embeddings are learned. All reviewers agreed that this is a novel area, and this work is a useful step toward more robust mitigation of indirect biases. Currently, the reviewers have concerns about the very constrained mitigation method, and whether it could be extended for biases other than gender.   There are also other mild empirical concerns around the mixed results. The authors’ responses helped to some extent. However, after reading the paper and the authors’ responses, as well as reviewers’ questions, the concerns seem to outweigh the novel aspect introduced in this work. Given it tackles a societally important problem, it would be great to have a robust and relatively general approach to quantify/mitigate indirect biases. This work could be strengthened with a stronger quantitative part to make a stronger submission in the future.

**Summary Of Ac-Reviewer Meeting:**

N/A